# Nanofiltration of the Remaining Whey after Kefir Grains’ Cultivation

**DOI:** 10.3390/membranes12100993

**Published:** 2022-10-13

**Authors:** Marjana Simonič

**Affiliations:** Faculty of Chemistry and Chemical Engineering, University of Maribor, Smetanova 17, 2000 Maribor, Slovenia; marjana.simonic@um.si

**Keywords:** kefir grain cultivation, remaining whey, nanofiltration

## Abstract

Acid whey is derived from fresh cheese. The proteins were isolated by a monolithic ion-exchange column. The remaining whey fraction was used as a starter culture substrate of kefir grains. The aim of this work was, firstly, to study the possibility of column replacement by a UF membrane. If we succeeded, the concentrate would be used as a starter culture substrate of kefir grains. The second part of the research was to purify the remaining solution. The idea was to separate this solution to the permeate and the concentrate by nanofiltration. Further application of both filtration streams was tested as aqueous solutions or dried matter. Chemical and microbiological analyses were performed of both the permeate and the concentrate. The permeate analyses showed that lactose had been fully removed. The aqueous permeate was not stable, mainly due to an increase of total bacteria from 10^3^ to 10^6^ CFU/mL. Therefore, the permeate was spray-dried. The dry permeate was added to the moisture solution in different concentrations. The results showed that up to 0.5% of the dry permeate could be added to the moisturizing solution, with negligible changes in properties having the ability to inhibit acne growth. Anaerobic digestion of industrial sludge was performed with the addition of an aqueous concentrate, which showed improvement in anaerobic fermentation.

## 1. Introduction

Whey offers a significant source of highly searched proteins in the market of dietary supplements, a substrate for the cultivation of dairy starter cultures, and a possible source of energy [1]. To the best of our knowledge, there is a lack of studies on how to apply the remaining solution after deproteinization. More stringent environmental discharge rules, the increasing cost of freshwater, and the need for rapid treatment of dairy wastewater, motivate technologists to look for newer methods for better treatment [2]. Biofertilizer was produced in a one-step process within a biofilm reactor under anoxic conditions. A beneficial practice is using whey on farming land as manure; however, it can have an adverse impact on soil, leading to decreased yields of crops, and resulting in severe groundwater pollution problems [1].

Reuse of waste solutions generated after different treatment processes is recognized in most water-scarce countries, e.g., deproteinized whey represents waste; instead of discharging such effluents into rivers and streams, nanofiltration (NF) membranes could be used to separate the feed into the permeate and the concentrate, and both could be reused as a secondary raw material [3]. The deproteinized whey could be used as a substrate for a starter culture, e.g., for kefir grains. Kefir grains are a mixed microbial consortium, a complex microflora that contains more than a hundred species of yeast and bacteria securely embedded in a physical structure [4]. This mixed microbiota is an example of a mutually beneficial community with lactic acid bacteria (LAB). Recent studies reported promising results, also related to antiviral effects, including in the case of COVID-19 [5]. The use of natural or genetically engineered microbial consortia presents advantages over individual strains, and is imperative for the designing of microbial consortia for biotechnological or medical purposes [6]. The probiotic properties of the strains are not affected by the growth temperature [7]. A certain lactic acid level could inhibit the growth of *Propionibacterium acne* [8].

It is well known that pretreatment prior to NF is essential for maintaining efficiency and protecting the membranes’ functions. The use of NF membranes largely reduces the chemical consumption when membrane separation is coupled with chemical treatment, due to the capability of the NF membranes in removing dissolved solids [9].

The quality and usage of the NF concentrate might be problematic, and is generally characterized by high levels of inorganic and organic substances, a low biodegradation, and potential ecotoxicity [10]. The anaerobic treatment of cheesy whey was performed using an acclimatized granular sludge in moderately acidic conditions [11]. Less average production of biogas was measured, but the amount of required NaOH for cheesy whey neutralization was lowered by up to 68%. In addition, anaerobic digestion as a promising process to solve the problem of cheesy whey disposal in small-to-medium enterprises, as well as the economic viability of the process, was proven [12]. In the anaerobic biodegradability analysis, methane yields reached 0.51 to 0.60 L CH_4_/g VS, which represented electrical and caloric potentials of 54 and 108 kW h/m^3^ for cheesy whey, respectively. Organic matter removal in all experiments was above 83%. Anaerobic digestion could be difficult because of the low pH (3–3.5) and the high protein and lactose contents, all of which inhibit methanogenic micro-organisms [13]. The process was improved by co-digestion with manure, or a mixture of food waste, cheesy whey, and olive mill wastewater [14]. The last combination improved biogas production by up to 2.7 times.

The main objective of the present study was to investigate the possibilities of the application of the remaining whey used for the cultivation of kefir grains after the removal of proteins. They were isolated by a monolithic ion-exchange column. Firstly, the ultrafiltration membrane was tested for deproteinization efficiency. The deproteinized whey is then used for the cultivation of kefir grains, and the remaining solution represented the feed into the nanofiltration unit. The permeate and the concentrate were analysed after nanofiltration. The research was performed on the possibilities of using the permeate as an additive in a cosmetic product, such as a moisturizing solution, and biogas production from the concentrate. No similar approach for spent whey treatment was found, despite numerous research groups working globally on dairy waste management [2].

## 2. Materials and Methods

### 2.1. Filtration Setup and Procedure

Proteins were isolated by monolithic ion-exchange chromatography from acid whey (Dairy Celeia, Petrovče, Slovenia), which was used as a substrate for the cultivation of probiotics. The first part of our research focused on the replacement of the existing ion-exchange chromatography protein isolation method with ultrafiltration (UF). A ceramic asymmetric multichannel alumina/zirconia membrane was chosen with a pore diameter of 0.05 μm in a tangential filtration system (JIUWU HI-TECH, Pukou, Nanjing, Jiangsu, China). The membrane was tested in a pilot system, self-made by the company Arhel Ltd. (Komenda, Slovenia) with a similar flowsheet as presented in Figure 1. The UF system feed was untreated acid whey in a 20 L tank with TMP from 0.5 to 2 bar.

The feed samples and the permeate were taken at the end of the trial, and subsequently analyzed. The whey flux was measured after every repeated trial. After every repeated trial with whey UF, the water flux was measured for fouling calculations.

NF was performed in the second part of our research. The remaining fraction after deproteinization was further nanofiltered in the MemCell, Osmo membrane system (Korntal-Munchingen, Germany) shown in Figure 1. The major difference between the systems for UF and NF was in the volume of treated samples and the module configuration: for UF membrane multichannel configuration, while plate membranes were applied for NF. The volume of the feed tank was 2 L for NF. The NF feed solution represented the remaining whey from the cultivation of the probiotics (RW). It flowed from the feed tank through the valve and the pump to the membrane cell. The system was equipped with flow meters (F) and pressure meters (P). The permeate was withdrawn from the system continuously, and the concentrate was returned to the feed tank. The transmembrane pressure (*TMP*) was kept at 25 bar for the NF membrane, based on a preliminary study where the water flux was highest [15]. Each trial was performed at a constant temperature of 4 °C using a thermostat, and repeated three times. The feed (RW), the permeate, and the concentrate samples were taken at the end of the trial and subsequently analyzed.

Multi-Channel High flux UF Ceramic membrane (Jiangsu Jiuwu Hitech Co., Ltd., Jiangsu, China) and flat-sheet NF membrane (Suez, Auburn, AL, USA) were used, containing the properties listed in Table 1. The UF membrane area was 0.418 m^2^, and the NF membrane area was 0.008 m^2^.

### 2.2. Zeta Potential Measurements

The zeta potential was measured using a special measuring cell for ceramic membranes within an electrokinetic analyzer (SurPASS^TM^3, Anton Paar GmbH, Graz, Austria). The membranes were wetted with a 0.001 M KCl solution, which was also used as a background electrolyte. 0.1 M NaOH was used as the titration liquid for the determination of the zeta potential on the pH dependence in the range from pH = 3 to pH = 9. The zeta potential was calculated from the measured streaming flow using the Helmholtz–Smoluchowski equation, as in our previous work [15].

### 2.3. SEM Imaging

The UF membrane was examined with an FEI, SIRION-400 Field Emission Scanning Electron Microscope (FESEM).

### 2.4. NF Membrane Characterisation

The NF membrane was examined with a Phillips XL-30 Scanning Electron Microscope (SEM). 

### 2.5. Chemical Analyses

The chemical analyses of the feed and the permeate were performed according to ISO standards in three replicates. The standard methods are summarized in Table 2. The analyses were chosen in accordance with the Slovene legislation for dairy emissions into the environment [16]. An important parameter among the general parameters is dry matter (*DM*). Additionally, the Elemental Analyzer PerkinElmer Series II 2400 was used to determine the carbon and nitrogen (in order to determine the C/N ratio) contents in the concentrate after nanofiltration and the digestate.

The volatile solids content (*VS*) amount was determined in the digestate according to SIST EN 15169 (2007).

### 2.6. Microbiological Analyses

The standard methods used for microbiological analyses, which were performed for the aqueous feed, the permeate, and the concentrate, are seen in Table 3.

### 2.7. Spray-Drying of the Permeate

The drying of the suspension with the starter culture was carried out in a Spray Dryer (Büchi mini-B-290, Büchi Labortechnik AG, Flawil, Switzerland) equipped with a nozzle type of 0.7 mm. The suspensions were fed into the chamber through a peristaltic pump at a constant flow rate and at an inlet air temperature adjusted to 150 ± 3 °C. The outlet air temperature was adjusted to 72 ± 1 °C. The aspiration was maintained at 100%; the drying air flow rate was kept at 32 m^3^/h; the air pressure was 0.6 MPa; and the feed flow was 10 mL/min. The dried powders were collected and stored under low vacuum conditions.

Experiments with mixing spray-dried permeate and moisture solution were performed, and the viscosity, using a Cannon-Fenske Viscometer (Zematra BV, Antwerp, The Netherlands), was measured after 24 h.

### 2.8. Anaerobic Fermentation

The mono- and co-digestion experiments with nanofiltration concentrate were conducted in 0.25 L batch reactors (~180 mL of working volume) and a retention time of 30 days. The reaction mixtures were prepared in two parallels, with the inoculum/substrate ratio of 2:1 and dry matter content of 4.5 wt.%. The inoculum/substrate ratio of 1:1 was applied in co-digestion. The inoculum was taken from a biogas plant-treating poultry manure and energy crops (Draženci, Slovenia). The mixtures contained 2.4 g of the substrate and 4.8 g of the inoculum on a dry basis, and a buffer solution was added to dilute the reaction mixtures to the selected dry matter content [17].

The volume of biogas produced was determined volumetrically by the water displacement method [17]. The methane share was determined by an Optima 7 biogas analyzer.

### 2.9. Calculations

Flux *J* ((L/m^2^·h) was determined as flow *q* ((L/min)) divided by membrane area *A* (m^2^) according to Equation (1):*J* =*q*/*A*(1)

Reversible fouling *F*_r_ (-) was determined according to Equation (2):*F*_r_ = (*J*_w_ − *J*_s_)/*J*_w_(2)

Millipore water flux through the virgin membrane *J*_w_ was measured at 2 bar. The second step was sample flux *J*_s_ measured at the same *TMP*. After the sample nanofiltration, Millipore water flux through the membrane *J*_wf_ was measured again.

Irreversible fouling *F*_ir_ (-) was determined according to Equation (3):*F*_ir_ = (*J*_w_ − *J*_wf_)/*J*_w_(3)
where

*J*_w_ = Millipore water flux through the virgin membrane (L/(m^2^h))

*J*_wf_ = Millipore water flux through the membrane after sample nanofiltration (L/(m^2^h))

*J*_s_ = sample flux (L/(m^2^h))

The total resistance *R*_t_ (m^−1^) was calculated according to Equation (4):*R*_t_ = *TMP*/(*η*·*J*)(4)
where

*η* = permeate viscosity (Pa·s), measured at company (Dairy Celeia, Petrovče, Slovenia).

The membrane resistance *R*_m_ (m^−1^) was calculated according to Equation (5):*R*_m_ = *TMP*/(*η_w_*·*J_w_*)(5)
where

*η_w_* = water viscosity (10^−3^ Pa·s at 20 °C)

During filtration the reversible (*R*_rev,_ m^−1^) and irreversible resistances (*R*_irr,_ m^−1^) were calculated by Equations (6) and (7):*R*_t_ = *R*_m_ + *R*_irr_ + *R*_rev_(6)
*J*_wf_ = *TMP*/(*η_w_*·*R_irr_*)(7)

Rejection (*R*) was calculated according to Equation (8):*R* = 1 − *c*_p_/*c*_f_(8)
where

*c*_p_ = concentration of component in the permeate

*c*_f_ = concentration of component in the feed.

## 3. Results and Discussion

### 3.1. Filtration with a UF Membrane

The most favourable *TMP* was determined based on the Millipore water flux measurement. The results of flux dependence on *TMP* are shown in Figure 2. The highest permeability was achieved at 1 bar and, at higher *TMP*, the flux tended to decrease; therefore, further tests were conducted at 1 bar.

In the next step, the flux of whey was measured at different TMP. The results are presented in Figure 3. A similar shape of curve was determined when filtering Millipore water; however, drastically lower fluxes were measured. A back flush had to be performed after every 15 min.

Flux measurements for whey and those of Millipore water before and after ultrafiltration at 1 bar are seen in Figure 4a,b. Figure 4b presents the flux of whey separately, because it was so low.

A drastic decrease of Millipore flux was observed after UF of whey due to fouling. The measurements of the total proteins in the concentrate confirmed that the membrane was fouled by the proteins. The isoelectric point (IEP) of whey was measured at pH = 3.6. We measured the whey pH, and it was 3.8, which is very close to the IEP. The fouling might be most severe in the IEP region due to weak electrostatic forces [18]. Therefore, the pH of whey was adjusted to 4.1, where the IEP of the membrane was determined, but the results did not improve much, and remained similar to those presented in Figure 4. The calculations of the fouling indicated 94% of irreversible fouling, and, thus, the flux of whey was low. Filtration resistance caused by reversible fouling (*R*_rev_) was calculated at 2.8 × 10^11^ m^−1^ and irreversible fouling (*R*_irr_) at 2.4 × 10^10^ m^−1^. The membrane had to be cleaned chemically. After the cleaning, which is presented in the next paragraph, the results were not improved, neither by changing the pH of the whey nor by changing the *TMP*; therefore, we concluded that the tested UF membrane was not a good replacement option for deproteinization of the acid whey, contrary to another study [18]. The worse membrane performance was attributed to the membrane fabrication procedure in accordance with the results of another study [19].

During UF of the whey, the flux was low, and remained low, which is in contrast to the common behaviour of whey nanofiltration, connected mainly to concentration polarization [20]. The authors found that, in the beginning, the decrease was high, then it slowed down and reached a stationary region. In our experiments, it seemed that polarization was not the major fouling mechanism; therefore, further analyses were made by measuring the zeta potential and SEM imaging.

### 3.2. UF Membrane Cleaning

In whey, after cultivation, different concentrations of proteins, lactic acid, lactose, and minerals (Ca, K, Fe) are present, which contribute to the fouling on the membrane’s surface.

After 200 min of working, the membrane fouled (not shown in the paper), because there was no flux, and cleaning was necessary. The cleaning agents were among a wide variety of chemicals, including acids (HCl, HNO_3_), a base (NaOH, KOH), enzymatic cleaners (Ultrasil 67), and their combination. To achieve high cleaning efficiency, the effects of physical factors were studied (velocity, temperature, and time). The results showed that the two stages, caustic and acetic cleaning including Ultrasil 69-Ultrasil 67 followed by 1% HNO_3_ acid, provided an effective recovery of 79%. The cleaning protocol was as follows: Rinsing for 10 min with Millipore water at 20 °C, caustic cleaning (1.5% Ultrasil 69) for 10 min at 55 °C, followed by 10 min cleaning with 4% Ultrasil 67 at 55 °C, rinsing for 10 min at 20 °C, HNO_3_ cleaning for 10 min at 55 °C, and rinsing until the neutral solution pH was reached. *TMP* during the cleaning operation was 0.5 bar. Since we could not disassemble the module (we only had this one), after finishing the whole research, the membrane surface was observed through SEM imaging. From Figure 5, it can be seen that Ca compounds dominated on the membrane’s surface. The SEM analysis was made at four different sites on the membrane’s surface. The results showed that mostly Ca remained on the surface. The cake layer forms by the deposition of material on the membrane’s surface rather than by penetration, in accordance with the SEM image (Figure 5). As seen from Table 4, the Ca content was found on the surface, while O, Al, and Zr were typical elements in the ceramic membrane whose properties are listed in Table 1.

The zeta potential of the ceramic membrane was measured. The ceramic membrane was cut transversely with a special saw. The results are shown in Figure 6. The isoelectric point was determined at pH = 3.6. The fouling layer on the membrane showed a strong buffering effect in pH 5–8. Titration was indicated slowly by many points in this pH region. The new membrane showed an isoelectric point at a higher value of pH = 5.1. Therefore, we could conclude that the fouling material has a negative charge. As is widely known, proteins have a negative charge which most probably fouls the membrane. Complete blocking and cake formation were the predominant mechanisms for this membrane, as also established in another whey UF study [21]. A lower buffering effect was detected by the zeta potential of the new membrane.

The zeta potential measurements for the monolithic membranes were not possible due to the need for the different measurement cells for the SurPASS^TM^ 3 Instrument, which were not available. Therefore, for the experiment with the nanofiltration membrane, the deproteinized whey was chosen after the monolithic membrane (RW).

### 3.3. Filtration with an NF Membrane

Based on the previous *TMP* measurements, the optimum pressure was chosen at 20 bar [14] for nanofiltration of the remaining RW. The fluxes of the nanofiltration of the RW and Millipore water before and after the nanofiltration of the RW are presented in Figure 7.

The fluxes of the filtered deproteinized whey were very similar to those reported for flow-through [15]. Similarly, the fouling was attributed to inorganic compounds, while large concentrations of Ca, K, and Fe ions remained in the deproteinized whey and fouled the NF membrane.

The zeta potential measurements and fouling study were reported in our previous paper [15]. With the nanofiltration of whey after the kefir grains‘ cultivation, very similar results and observations were obtained, and, therefore, are not presented herein.

The SEM analysis of the NF membrane was performed. The results are shown in Figure 8a,b. The top dense skin layer is seen in Figure 8a. The surface of the virgin NF membrane is inhomogeneous as reported in the other study [22]. On the surface of the fouled membrane in Figure 8b, some deposits can be seen, indicating organic matter, such as lactose, lactic acid, etc., from the deproteinized whey. Figure 8b shows a very low amount of the deposit, which is in agreement with the constant flux of deproteinized whey shown in Figure 7.

#### 3.3.1. Chemical Analyses

After the filtration of all samples, the parameters listed in Table 2 were measured in the RW samples. The results are gathered in Table 5.

The lactose was not found in the permeate, so it could be fully separated from the lactic acid. However, the lactic acid was distributed in the permeate and the concentrate, and could not be separated between these two phases. Furthermore, the results showed that lactic acid remained in both the permeate and the concentrate, and could not be separated with NF, which is consistent with the literature reporting 54–57% removal at acidic pH [22].

#### 3.3.2. Microbiological Analyses

Next, microbiological analyses were performed in order to explain the content of the chemical parameters. The results are gathered in Table 6.

Therefore, we tried to separate the lactose and lactic acid with septic conditions. However, it was very hard to maintain such conditions with the available equipment. The temperature rose very quickly above 5 °C and, in all the experiments, the results were not improved. The lactic acid was always found in the permeate and the retentate, together with the increased colony-forming unit (CFU) of the total number of micro-organisms. The reason could be sought in the symbiotic interactions between the micro-organisms of the whey consortium, as well as in a substrate–inoculum synergistic relationship [1]. In addition, the lactic bacteria were studied in detail in another study [23]. Thus, it was not the aim of the present study to discuss such interactions in more detail. To overcome this problem, the permeate and the concentrate were treated further, each separately.

#### 3.3.3. Experiments with a Moisturizing Solution

A different mass of spray-dried permeate was added to the moisturizing solution and mixed for 24 h. After 24 h, the viscosity was measured (at 25 °C) and compared with the initial solution. The viscosity of the moisturizing solution remained the same measurement at 0.890 × 10^−3^ Pas. In a 0.1% solution, the viscosity was measured at 0.893 × 10^−3^ Pas, and remained the same. In a 0.5% solution, the initial viscosity was 0.928 × 10^−3^ Pa, and, after 24 h, increased to 0.9285 × 10^−3^ Pas. The solution with 1% added spray-dried permeate became turbid instantly, and we did not measure the viscosity. It was established that, due to the formation of precipitation, it was not appropriate for cosmetic use. It was concluded that a 0.5% addition is the maximum dose which does change the physical properties of the moisturizing solution negligibly. Other authors determined that the addition of whey prevents acne growth [8]. The measurements of the viscosity were repeated after each day. If the solution is kept in the fridge and mixed, the viscosity remains similar for up to one month.

#### 3.3.4. Experiments with Concentrate

The anaerobic digestion of the mono-digestion of the RW, the sludge of the vegetable oil industry (SVOI), and co-digestion of both materials were performed. The RW is problematic due to its low pH, but, in co-digestion with organic material, the pH is adjusted closer to neutral, and, therefore, it becomes more suitable for biogas production [11]. The acid pH of 3.7 would hinder biogas production; therefore, the pH was adjusted to 7. The SVOI was chosen based on claims that the SVOI in co-digestion with the RW improved biogas production [14]. The dry matter content in the SVOI was 37 g per 100 g, and in the RW 3.3 g per 100 g. In the first few days, low biogas production was reported [13]. Therefore, in Table 7, biogas production is shown after five days and followed up for one month (30 days).

Table 7 showed promising results. The co-digestion of the SVOI and RW concentrate was successful. As seen in Table 7, an amount of 1132 mL/(gVS) biogas was produced, while a very similar amount of 1172 mL/(gVS) was found of cheese whey co-digestion with olive mill wastewater [24]. The results for the RW showed relatively high biogas production at 650 mL/(gVS), while biogas production was reported for cheesy whey at 216 mL/(gVS) [25]. The improvement could be attributed to the elimination of lactose after nanofiltration of the RW, which has a high inhibitory effect on biogas production [26]. The RW concentrate after NF, unlike untreated cheese whey, has a fat content which is concentrated after nanofiltration at such a level that it could be attributed to higher biogas production. The results from Table 7 confirm such claims in accordance with the literature [13]. The methane content was 71% in the SVOI and 65% in the mixture of the SVOI with the RW. The percentage is in accordance with the literature, where 71% methane content was reported in the mixture of the sludge from the vegetable oil industry and the sewage sludge [27]. The C/N ratios 21/1 were determined in the RW concentrate and 18/1 in the SVOI + RW co-digestate. The ratio was very close to the ratio of 25/1 which was reported as the optimal value for anaerobic microbial activities [28]. Water recovered from the digestate is recycled in the anaerobic digestion unit, while dewatered digestate with DM content > 23 wt % is used as a soil enhancer in accordance with the literature [17].

## 4. Conclusions

The idea of the paper was to exploit the deproteinated whey fully after the cultivation of kefir grains. The study shows that the remaining whey fraction could be exploited fully by applying nanofiltration to separate useful compounds from the remaining whey solutions after kefir grain cultivation. Chemical and microbiological analyses were performed of the remaining whey, permeate and concentrate. The permeate was included successfully into the moisturizing solution after spray-drying, while the aqueous permeate was unstable. Up to 0.5% of the dry permeate could be added to the moisturizing solution without noticeable changes in its properties. Anaerobic co-digestion of the NF concentrate and the sludge from the vegetable oil industry showed that biogas production is quite satisfactory, with 1132 mL/(gVS) produced. Around 71% of the produced biogas was composed of methane.

## Figures and Tables

**Figure 1 membranes-12-00993-f001:**
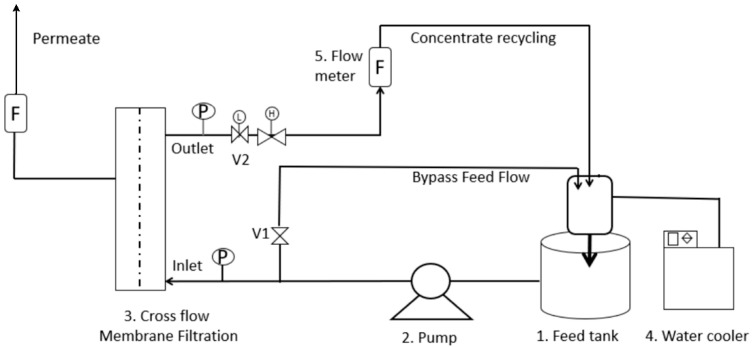
Schematic diagram of NF process.

**Figure 2 membranes-12-00993-f002:**
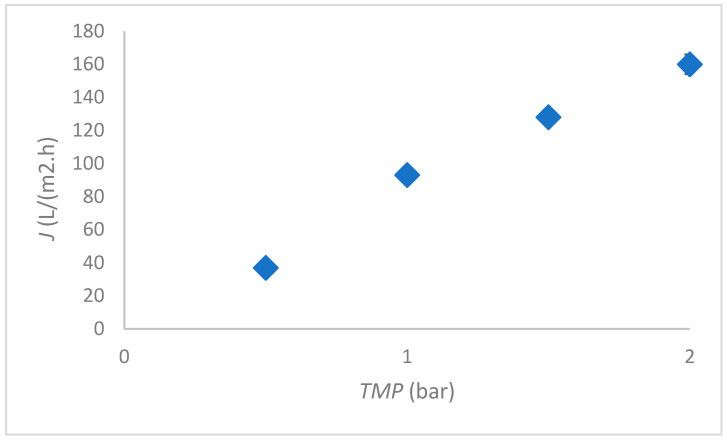
Flux dependent on *TMP* for the ceramic membrane.

**Figure 3 membranes-12-00993-f003:**
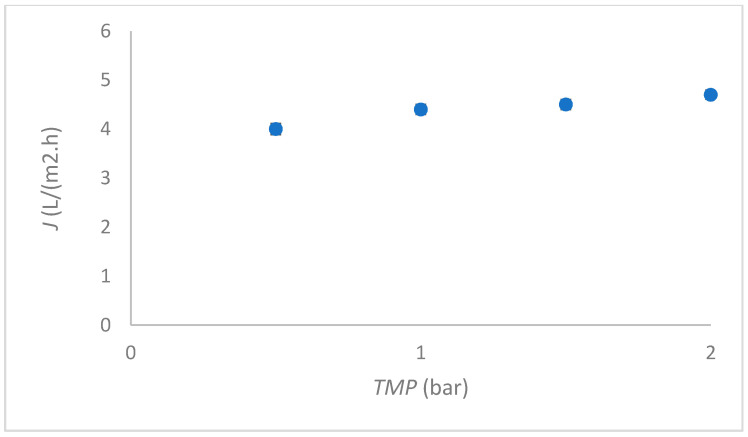
Flux dependent on *TMP* for the ceramic membrane at filtering whey.

**Figure 4 membranes-12-00993-f004:**
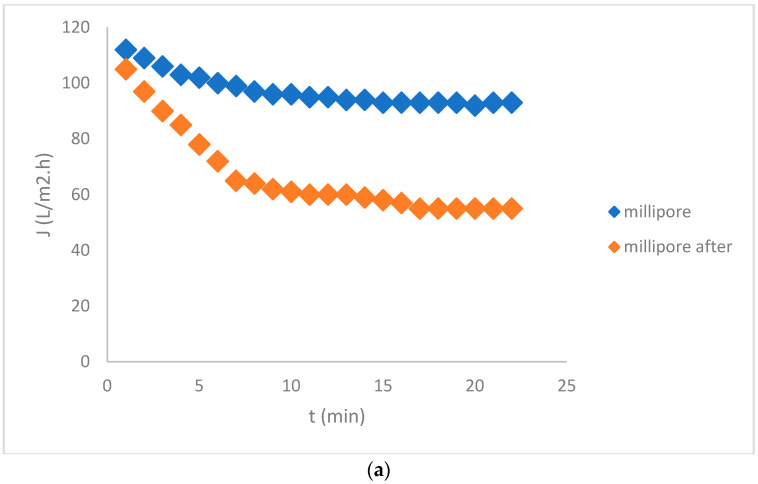
Flux dependent on time for the ceramic membrane, (**a**) For Millipore water before and after whey filtration and (**b**) Whey.

**Figure 5 membranes-12-00993-f005:**
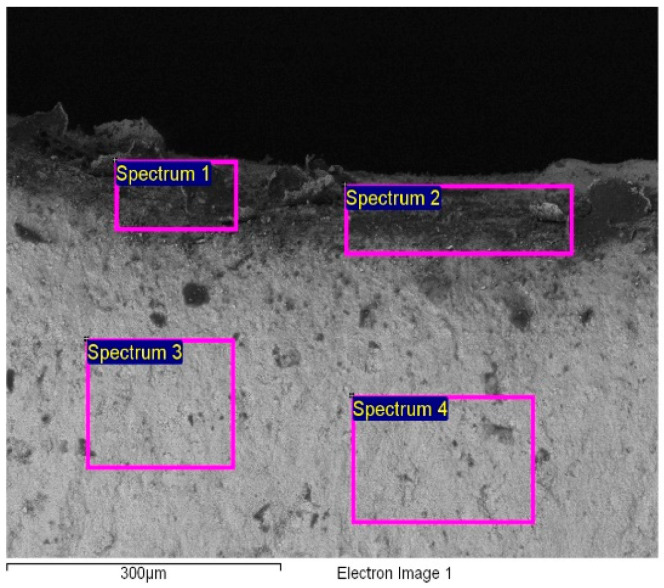
SEM micrograph of the ceramic membrane after UF.

**Figure 6 membranes-12-00993-f006:**
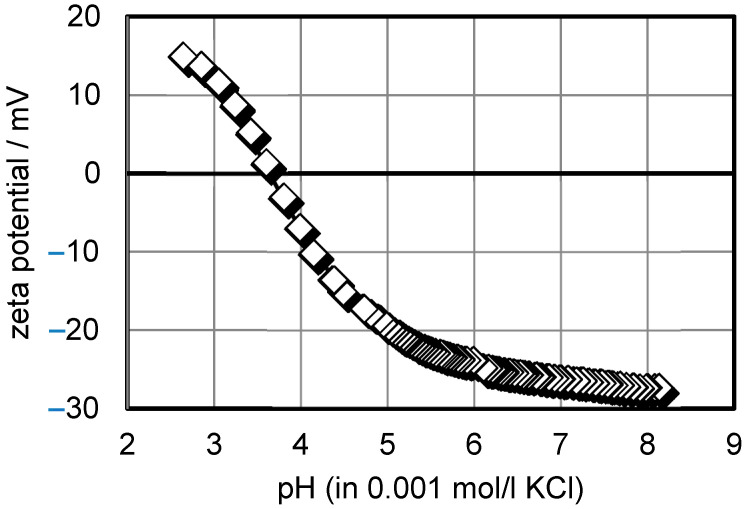
The membrane zeta potential measurement.

**Figure 7 membranes-12-00993-f007:**
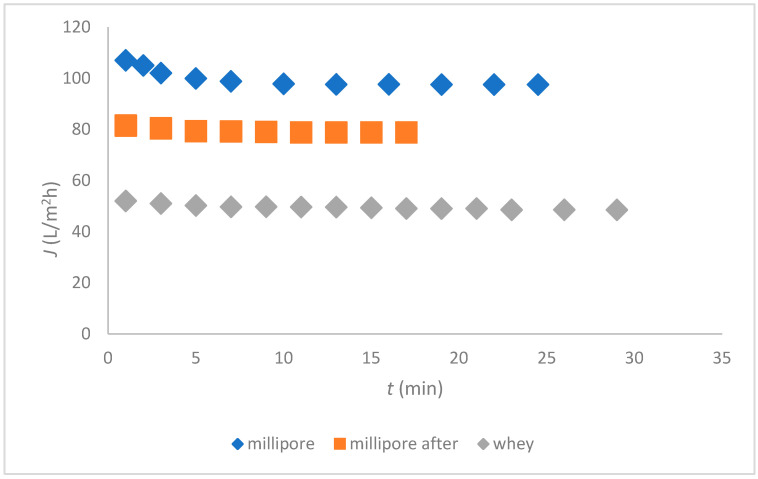
Flux dependent on time for the NF membrane.

**Figure 8 membranes-12-00993-f008:**
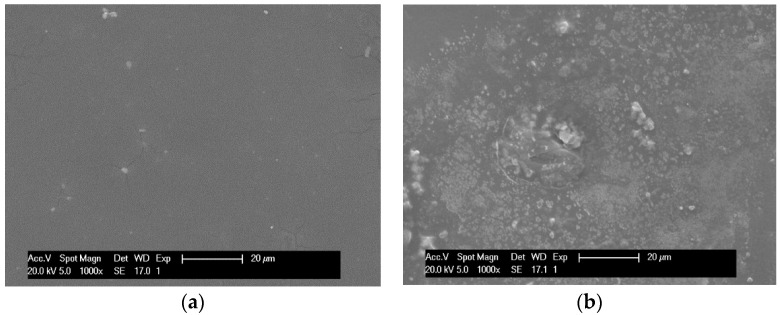
SEM micrographs of (**a**) virgin NF and (**b**) fouled NF membrane.

**Table 1 membranes-12-00993-t001:** The UF and NF membranes’ characteristics.

Parameter	UF	NF
Producer	Jiuwu Ceramic membrane	Suez
*P*	0.5–10 bar	0.5–28 bar
*T*max	100 °C	50 °C
pH	0–14	2–11
Pore size	50 nm	10 nm
Morphology	Ceramic multichannel α-Al_2_O_3_/ZrO_2_	Thin film poly(piperazineamide)

**Table 2 membranes-12-00993-t002:** The standard methods used for the chemical analyses.

Parameter	Units	Standard Method
*w* (protein)	g/100 g	ISO 8968/IDF 30-3 (2004)
*w* (lactose)	g/100 g	ISO 22662/IDF 198 (2007)
*w* (fat)	g/100 g	ISO 1211/IDF 1 (2010)
*DM*	g/100 g	ISO 6731/IDF 21 (2010, mod.)
*w* (Lactic acid)	mg/100 g	Boeringer Mannheim/R-Biopharm kit, 2021
*w* (ash)	g/100 g	AOAC 938.08, gravimetric
*w* (Ca)	g/100 g	AOAC 938.08, gravimetric
*w* (K)	g/100 g	AOAC 938.08, gravimetric
*w* (Fe) pH	mg/100 g -	AOAC 938.08, gravimetric ISO 10523 (1996)

**Table 3 membranes-12-00993-t003:** The standard methods used for microbiological analyses.

Parameter	Standard Method
*E.coli*	ISO 16649-2 (2001)
*Coliforms*	ISO 4832 (2006)
*Listeria monocytogenes*	ISO 11290-1 (2017)
*Salmonella* spp.	ISO 6579-1 (2017)
*Total number of microorganisms*	EN ISO 4833-1 (2013)
*Staflococcus aureus*	ISO 6888-2 (1999)

**Table 4 membranes-12-00993-t004:** SEM analyses after UF.

Spectrum	O	Al	Ca	Zr
1	51.82	10.97	1.64	35.56
2	51.30	10.7	0.68	37.32
3	30.97	1.00	-	68.03
4	30.86	1.00	-	68.14

**Table 5 membranes-12-00993-t005:** Measured chemical parameters after nanofiltration.

PARAMETER	RW	Permeate	Rejection (%)
*w* (proteins) (g/100 g)	0.69	0.04	94
*w* (lactose) (g/100 g)	0.428	0.0	100
*w* (fat) (g/100 g)	0	0	-
*DM* (g/100 g)	6.02	0.38	94
pH	3.3	3.1	3.2
*w* (lactic acid) (mg/100 g)	661.1	316.4	50
*w* (ash) (g/100 g)	0.921	0.171	81
*w* (Ca) (g/100 g)	0.108	0.051	53
*w* (K) (g/100 g)	0.186	0.051	73
*w* (Fe) (mg/100 g)	0.120	0.034	72

**Table 6 membranes-12-00993-t006:** Microbiological analyses after nanofiltration.

PARAMETER	RW	Permeate	Concentrate
*E. coli*	<1 CFU/mL	<1 CFU/mL	<1 CFU/mL
*Coliforms*	<1 CFU/mL	<1 CFU/mL	<1 CFU/mL
*Listeria monocytogenes*	ND *	ND	ND
*Salmonella* spp.	ND	ND	ND
*Total number of microorganisms*	1600 CFU/mL	9.4 × 10^5^ CFU/mL	1.1 × 10^6^ CFU/mL
*Staphylococcus aureus*	<10 CFU/mL	<10 CFU/mL	<10 CFU/mL

ND * not detected.

**Table 7 membranes-12-00993-t007:** Biogas production.

Day/Substrate	5	10 Biogas (mL/(gVS))	20	30
SVOI	558	1260	1513	1630
RW concentrate	340	597	640	650
SVOI and RW concentrate	450	930	1054	1132

## Data Availability

Not applicable.

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
