# Peer review of "Nanofiltration of the Remaining Whey after Kefir Grains’ Cultivation"

_membranes, 2022, doi:10.3390/membranes12100993_

Round 1

Reviewer 1 Report (Previous Reviewer 2)

The author considered all the points indicated in the review, and the new version is considerable improved compared to the original manuscript.

Author Response

Response: Thank You for Your kind revision.

Reviewer 2 Report (Previous Reviewer 3)

I noticed an improvement in the manuscript. The author replied satisfactorily to all my requests.

I consider that the manuscript can be published in this form.

Author Response

Thank You for Your kind revision.

Reviewer 3 Report (New Reviewer)

this work is valuable  

Author Response

I thank Reviewer for revision. Responses are found below Reviewer comment.

This article aims to study the possibility of column replacement by a UF membrane. The research idea was to separate remaining whey after kefir grains` cultivation permeate and concentrate by nanofiltration. However, there are several issues that should be stressed before it is considered for publication. This is an average level work and I recommend it for publication after doing some minor revisions.
.The Comments are addressed below.

Comments:

Comment 1: abstract

  • This part is adjustable and good written

Comment 2:   Introduction part

  • This part is adjustable it covers background of the related work up to now  

    Comment 3: Materials and Methods part

2.1. Filtration setup and procedure

- This part need to be clearer, are you work with a batch set-up test first with UF membrane then the output from UF you collect it for testing with NF membrane with the same set-up?

Response: The fraction after deproteinization with UF was further treated with NF system. Please see lines 93-95. The principle of the UF and NF is similar as presented in Figure 1, but there were two different set-ups for each filtration performance: please see lines 87-95.

Comment 4:   Result, Discussions, conclusion part:

  • This part is adjustable and good written

Comment 5: References:

  • The refs. Must be adjustable as the journal format required. Please recheck the "Reference" section, carefully, according to the journal's format

Response: The references were carefully re-checked as suggested and changes were made using “Track Changes”.

Comment 6:

Please submit a "Plagiarism Detection Software" report (for instance iThenticate) of your manuscript with the revised draft as a supplementary file.

Response: The report is provided as suggested as a supplementary file.

This manuscript is a resubmission of an earlier submission. The following is a list of the peer review reports and author responses from that submission.

Round 1

Reviewer 1 Report

I hardly see any significant improvement in the manuscript. The author replied that he/she has revised the manuscript, but such changes are not significant at all for this paper to be published in Q1 journal.

Instead of further analyzing why the membrane exhibited EXTREMELY low flux, the author just simply ignored the factors behind the low flux (Figure 4a). This is completely irresponsible.

The author further answered that “it was impossible to increase the flux, therefore it was concluded that the membrane chosen was not suitable”. If the membrane is not suitable, why he/she chose the membrane at the first place. It is not logic at all.

When I asked about why rejection data is missing, the author replied - The rejection of proteins was 80 %, which is much lower than reported up to 99 % rejection. The author did not provide a satisfactory reply.

When I asked the author to provide FESEM image of NF membrane, he/she seemed does not understand the comment. FESEM image was requested, but the author provided other kind of analyses.

When I asked about why microorganisms could permeate through NF membrane and be detected in permeate (Table 6), I did not see any statement in Section 3.3.1. to justify this (even though the author said so).

All the figures are not professional prepared. These comments are already given in the first review and yet no action was taken by the author.

Author should not submit this kind of manuscript to embarrass him/herself.

Reviewer 2 Report

The manuscript was improved.
